# Feline Lymphoma: Patient Characteristics and Response Outcome of the COP-Protocol in Cats with Malignant Lymphoma in The Netherlands

**DOI:** 10.3390/ani13162667

**Published:** 2023-08-18

**Authors:** Hannah Versteegh, Maurice M. J. M. Zandvliet, Laurien R. Feenstra, Francine E. M. M. van der Steen, Erik Teske

**Affiliations:** Department of Clinical Sciences, Faculty of Veterinary Medicine, Utrecht University, Yalelaan 108, 3584 CS Utrecht, The Netherlands; m.zandvliet@uu.nl (M.M.J.M.Z.); l.r.feenstra@uu.nl (L.R.F.); f.e.m.m.vandersteen@uu.nl (F.E.M.M.v.d.S.); e.teske@uu.nl (E.T.)

**Keywords:** feline, lymphoma, FeLV/FIV, anatomical types, chemotherapy

## Abstract

**Simple Summary:**

Recent data suggest that feline lymphoma may no longer be strongly associated with viruses like feline leukemia virus (FeLV) and feline immunodeficiency virus (FIV), possibly causing a change in the occurrence of specific anatomical subtypes and treatment outcomes. This study aimed to analyze this shift in disease presentation in a country with a low appearance of FIV/FeLV, such as the Netherlands, to investigate if these changes can be solely attributed to the prevalence of these viruses or if other risk factors should be considered. In total, 174 cats with large cell lymphoma (110 treated) during the last 10 years were included and compared to previous data from the Netherlands. This study demonstrates a shift in patient characteristics of cats with malignant lymphoma in the Netherlands, which could not be ascribed to differences in FIV/FeLV occurrence. While treatment response remained consistent, some anatomical subtypes showed longer disease-free periods.

**Abstract:**

Feline lymphoma is currently less commonly associated with retrovirus infections as the feline leukemia virus (FeLV) and feline immunodeficiency virus (FIV). This is thought to have caused a shift in the distribution of anatomical subtypes and eventually have led to poorer treatment outcomes. The aim of this study was to evaluate whether this change was also notable in the Netherlands, a country historically known for its low prevalence of FeLV and FIV, and to determine its consequences on treatment response. A 10-year cohort of 174 cats with large cell lymphoma (110 treated) were included and compared to historical data from previously published reports in the Netherlands. Of the 90 cats screened, only one tested positive for FeLV and three for FIV. The most current cohort had an increased age (median 8.7 years) and fever Siamese cats (6.3%) compared to previous reports, with alimentary (24.5%) and nasopharyngeal lymphoma (22.7%) being the most common subtypes. Sixty-six of the one hundred and ten cats (60%) went into complete remission, (CR) resulting in a median disease-free period (DFP) of 763 days, with nasopharyngeal and mediastinal having the longest DFP. The median overall survival time was 274 days with an estimated 1-year survival of 41.3% and a 2-year survival of 34.6%, respectively. Patient characteristics of cats with malignant lymphoma in the Netherlands have changed over the years, but this cannot be explained by differences in FeLV/FIV prevalence. Although the overall response rate to therapy did not change over time, for some lymphoma subtypes, longer DFPs were observed compared to 30 years ago.

## 1. Introduction

Malignant lymphoma is the most common hematopoietic malignancy in domestic cats [1,2,3], with an estimated incidence of about 100–200 cases per 100,000 cats [4]. The age of presentation in most studies appears to be bimodal, with a first peak at around 2 years of age and the second between 10–12 years [5]. The first peak consists primarily of young, male Siamese and Oriental short-hair cats, which seem to be predisposed compared to other purebred cats, and European domestic shorthair cats [5,6,7,8]. Possible reported risk factors for developing lymphoma are a positive feline leukemia virus (FeLV) and/or feline immunodeficiency virus (FIV) infection [7,9,10,11], environmental influences such as exposure to tobacco [12], radon, agricultural chemicals [13] and immunosuppressive diseases or treatments [7].

Especially the linkage to FeLV/FIV has been thoroughly studied [14,15,16,17,18]. FeLV-associated lymphomas seem to arise most of the time from the T-cell lineage, causing a mediastinal or peripheral form in often younger cats [7,14]. FIV-associated lymphomas, on the other hand, emerge from a B-cell origin [19] and develop in extra-nodal sites and older cats [3,19,20]. Over the last decades, a significant change has been seen in the prevalence of FeLV and FIV infections in multiple countries. Until the 1980s, about 70% of the cats with malignant lymphoma were FeLV positive [10,21,22]. However, since then, the use of effective testing, eradication programs, and vaccination against FeLV numbers have decreased significantly to 5–14.5% [6,7,8]. Although FeLV-associated lymphoma numbers have dropped, the overall incidence of lymphoma in cats does not seem to decrease much and might even have increased [8]. In particular, the alimentary and peripheral forms are seen more frequently these days, while the incidence of the mediastinal form has declined [7,23]. It has been stated that this so-called shift in the anatomic location of the lymphoma may have been caused by the decrease in FeLV and FIV infections. Some countries, such as the Netherlands and Australia, are known for their longstanding history of low FeLV/FIV infection rates, either by eradication programs or by geographic differences [24,25,26]. Given the historically limited role of retroviruses in feline lymphoma in these countries [25,27,28,29], it raises the question whether the beforementioned shifts in anatomical subtypes were observed there as well, and if so, this might support a role for other risk factors than retroviral infections.

Lymphoma can be divided according to its anatomic location, into an alimentary, mediastinal, peripheral, nasopharyngeal, cutaneous, abdominal, and miscellaneous form, the latter containing the remaining locations [30,31]. The preferred treatment for lymphoma exists of a multicyclic, multidrug chemotherapy protocol [21,23,25,29]. A variety of chemotherapeutic protocols have been reported, with a variation in response rates, disease-free periods (DFP), and overall survival times (OST). The COP-protocol (vincristine, cyclophosphamide, and prednisone) has been proven to be effective with response rates as high as 77% and 2 years survival rates of more than 46% [25,29,32]. The efficacy of the treatment seems to depend on the anatomical type of lymphoma, with alimentary and abdominal subtypes being poorly responsive and mediastinal highly responsive to chemotherapy [30,33]. As a result, one could speculate that a shift in the prevalence of the anatomical lymphoma subtype might therefore result in a difference in overall response rates.

As previously mentioned, the global decline in the prevalence of FeLV and FIV may have altered the clinical presentation of feline lymphoma and may therefore have influenced treatment outcomes. The objective of the present study is to test if similar changes in feline anatomical lymphoma subtypes have occurred in the Netherlands, a historically low FeLV prevalence region, as in the rest of the world, and if these changes have had an impact on the overall treatment outcome. Since this is a single institutional study, any changes observed are unlikely to be related to geographic variations, retroviral status, or variations in treatment protocols.

## 2. Materials and Methods

### 2.1. Animals

Medical records of all cats presented with malignant lymphoma at the Utrecht University Clinic for Companion Animals (UUCCA) between 2010 and 2020 were collected. Cats treated according to the modified COP-protocol established by Teske et al. (2014) were selected and compared to two groups of cats with malignant lymphoma treated at the UUCCA in earlier periods with the same protocol (1984–2002 and 2004–2010) [25,29]. The diagnosis of malignant lymphoma was confirmed in all cats by cytology of a fine-needle aspiration biopsy or histology of a biopsy of a diseased organ or an excised lymph node. In all three periods, all the investigated cases included were only large cell lymphomas. Small cell lymphoma (i.e., low grade) and large granular lymphocyte lymphomas were excluded.

### 2.2. Data Collection

Animals for this study were selected from the UUCCA patient management system (Vetware^®^ (TIS VetWare, Agfa HealthCare, Vienna, Austria), in use since 2004) from 2010 to 2021. Information obtained from this database included age, breed, gender, exact diagnosis, medical history, drug administration reports, and reports of clinical state and adverse events (AE; vomiting, diarrhea, anorexia, obstipation, fever, pain, hair loss, or dullness). As not all tumors in these cats were staged, stage was not included as part of the prognostic criteria study.

Screening for FeLV and FIV was performed randomly by the SNAP FIV/FeLV Combo test, which has a sensitivity of 93.5% and specificity of 100% for FIV and a sensitivity of 98.6% and specificity of 98.2% for FeLV according to the manufacturer (IDEXX Laboratories, Westbrook, ME, USA). All FeLV-positive test results were confirmed with a real-time PCR [34]. Positive FIV cases were confirmed by Western blot method [26].

The selected cats were classified based on the anatomic location of the lymphoma: peripheral nodal, mediastinal, nasopharyngeal, abdominal extra-alimentary, alimentary, cutaneous, and miscellaneous, including laryngeal, skeletal, and ocular forms [31].

### 2.3. Treatment

A modified version of the COP chemotherapy protocol described by Cotter et al. [10] was used for all cats [25,29]. During week 1, 2, 3, and 4, induction of remission was established with weekly dosages of vincristine, combined in week 1 and 4 with cyclophosphamide, followed by a maintenance phase using the simultaneous administration of vincristine and cyclophosphamide every 3 weeks till week 52. Prednisolone was installed on the first day and an oral dose of 50 mg/m^2^ was given daily. Hereafter, prednisolone was continued until relapse or for one year in remission. Vincristine and cyclophosphamide were administered simultaneously intraperitoneally (IP) as previously described [29]. Dosages were based on weight and varied between 0.6 mg/m^2^ for vincristine and 250 mg/m^2^ for cyclophosphamide. Prednisolone was given 1 mg/kg/day orally (PO).

### 2.4. Response Evaluation

Remission was determined on the combination of absence of clinical signs (history), physical examination, and, if appropriate, diagnostic imaging. For cats with mediastinal lymphoma, repeated thoracic radiographs were made; in renal and alimentary lymphoma, repeated abdominal ultrasonography was performed. Complete response (CR) was defined as 100% regression of measurable lesion(s) and clinical signs, partial response (PR) as less than 100% but more than 50% regression of disease, and partial improvement of clinical signs. When less than 50% of tumor volume reduction was achieved or when there was an increase of no more than 25%, this was stated as no change (NC). An increasement of more than 25% or the formation of new tumors was defined as progressive disease (PD). For nasopharyngeal lymphomas, CR was mainly based on the complete disappearance of clinical signs for at least one month.

### 2.5. Statistical Analysis

Data were tested for normality using the Kolmogorov–Smirnov test, after which non-parametric data were analyzed with the Kruskal–Wallis test to show differences among groups in case of interval data. Differences between groups in case of ordinal or ratio data were measured by using the chi-square test or Fisher’s exact test. Survival curves were calculated by the Kaplan–Meier product limit method. Disease specific survival time was calculated as the interval between the start of treatment until the date the cat was last known to be alive. Cats were censored in the survival analysis when cats were lost in follow-up (1), died from other causes than lymphoma or chemotherapy (2), or when they were alive at the time of analysis (3). The disease-free period (DFP) was calculated for cats that had obtained a complete remission and defined as the interval from start of treatment till relapse or to the date on which the cat was last known to be free of disease. For the DFP analysis cats were censored when they died in remission from causes unrelated to lymphoma or chemotherapy (1), were lost to follow-up (2), or were in complete remission at the time of analysis (3). Comparison between groups in the survival data were made using the log-rank test. A result was considered statistically significant if the *p*-value was <0.05. All statistical analyses were performed by using IBM Statistics SPSS 26^®^.

## 3. Results

### 3.1. Patient Characteristics

In total, 174 cats were included in this study, of which 110 animals were treated with the modified COP-protocol (Table 1). In order to prove that no selection bias for treatment was performed we compared the two groups to each other. Although the European shorthair and other mixed breeds were most often seen, other frequently affected breeds were the British Shorthair, the Siamese, and the Maine Coon. Male cats were seen more often in both the untreated group (62.5%) and the treated group (66.3%). The mean age of the untreated group was 9.1 years (range, 2–16 years) and of the treated group 8.8 years (range, 1–16 years) (*p* = 0.529). In both groups, cats with mediastinal lymphoma were significantly (*p* = 0.002) younger (mean age 3.0/4.4 years) compared to the cats with other anatomic subtypes (9.4/10.8 years), except for the peripheral nodal form (mean age 7.3/7.7 years). Of the 90 cats tested on FIV/FeLV, three were positive for FIV and one was positive for FeLV. Only one positive tested FIV cat was treated with chemotherapy. This cat was diagnosed with nasal lymphoma. The other FIV-positive cats were diagnosed with intestinal and nasal lymphoma, while the FeLV cat was diagnosed with a mediastinal form. The alimentary (27%) and nasopharyngeal (23.6%) forms were seen most often, while only 8% of the cats were diagnosed with mediastinal lymphoma.

### 3.2. Comparison of Different Time Periods

The current results were compared to the historical data of the two groups of cats with malignant lymphoma treated at the UUCCA between 1984–2002 and 2004–2010 [25,28]. To avoid bias, only the treated animals (*n* = 110) were used for this comparison. 

During 2011–2020, more cats with malignant lymphoma were treated with the COP protocol (*n* = 110) than in the studies of 1984–2002 (*n* = 61) and 2004–2010 (*n* = 26). The proportion of Siamese cats was significantly larger in the period 1984–2002 (26.2%) compared to the period 2004–2010 (3.8%; *p* = 0.016) and the period 2011–2020 (7.3%; *p* < 0.001). No significant difference in the frequency of this breed was noted between the last two periods. 

No significant gender difference was noted between the three periods. From 1984–2002, 65.6% of the cats were male, while 34.4% were female. From 2004–2010, 50% of the cats were male and 50% female. The neuter status changed significantly over the years. The proportion of intact cats was significantly higher in the period 1984–2002 (27.0%) compared to the periods 2004–2010 (7.7%; *p* = 0.037) and 2011–2020 (3.6%; *p* < 0.0001). No significant difference was present between the last two periods. The median age is currently significantly higher (median 9 years; mean age 8.7 years) compared to 30 years ago (median 7 years; mean age 6.8 years) (*p* = 0.006). In the period between 1984 and 2002, a clear bimodal age distribution was visible [25], while this distribution can no longer be demonstrated in the periods of 2004–2010 and 2011–2020 (Figure 1). The FeLV/FIV status did not change significantly over the three time periods.

A significant (*p* < 0.001) shift in anatomical subtype is found over the three time periods. Between 1984–2002, the most common form was the mediastinal type (36.1%). During 2004–2010, nasopharyngeal lymphoma (38.5%) was seen most often, while alimentary lymphoma is the most frequent type (24.5%) in the group of cats treated during 2011–2020. The decline in mediastinal lymphoma was not only due to a decrease in Siamese cats, as it was also significant in the non-Siamese breeds (*p* = 0.0145).

### 3.3. Response to Treatment and Survival Data

In total, 66 animals (60%) of the 110 cats treated with the COP protocol had a CR, 23 cats (20.9%) had a PR, 10 cats (9.1%) showed an SD, and 11 (10%) cats had PD. The CR rate differed significantly between the various anatomical subtypes (*p* = 0.003) (Table 2). The miscellaneous (84.6%) and the mediastinal (83.3%) had the highest CR rates, while the cutaneous (0%) and non-alimentary abdominal forms (29.4%) had the lowest CR rates.

The median DFP was 763 days (95% CI, 150–1376) with a 1-year estimated DFP of 58.9%, and a 2-year estimated DFP of 52.7%. (Figure 2) The median survival time was 274 days (95% CI, 179–387), with an estimated 1-year survival of 41.3% and a 2-year survival of 34.6%.

No significant difference (*p* = 0.105) was found for the median DFP between the different anatomical subtypes. Since none of the cutaneous forms had a CR, no DFP could be calculated (Table 2). The median survival time differed significantly (*p* = 0.019) between the different anatomical subtypes. The 1-year survival rate was highest for cats with mediastinal lymphoma (66.7%). Abdominal and peripheral lymphoma, on the other hand, had a 1-year survival of 17.6% and 10.1%, while none of the cutaneous lymphoma patients reached the 1-year survival (Figure 3).

Of the 27 alimentary lymphomas, 19 were intestinal (70%) and 8 were gastric (30%) lymphomas. The median DFPs for the gastric type (217 days) and the intestinal type (440 days) were not significantly different (*p* = 0.823), nor was the median survival (*p* = 0.727), with median survival times of 260 and 486 days, respectively.

### 3.4. Comparison Response to Treatment in Different Time Periods

No significant difference in overall response rate was seen in comparison to the earlier periods of 1984–2002 (*p* = 0.125) and 2004–2010 (*p* = 0.135) (Table 3). The number of cats per anatomical location was too small to analyze response rates for individual subtypes. The median DFP increased from 251 days (period 1984–2002) to 763 days (period 2011–2020), mainly attributed to a large increase in the median DFP for nasopharyngeal and mediastinal subtypes.

## 4. Discussion

In the earliest cohort, the population of cats diagnosed with lymphoma consisted to a large extent of young, intact, male cats, and included a high number of Siamese cats [5,7,9,35]. Currently, this group is less commonly presented, resulting in a population predominantly consisting of older castrated European shorthairs [7,23,25]. At the same time, a shift occurred in the proportion of anatomical subtypes. The most common mentioned reason for these developments has been suggested to be the worldwide decline in FeLV/FIV prevalence. However, this study fails to support this assumption and suggests that other risk factors should be considered.

The results of this study demonstrated that cats currently present at an older age compared to 20–30 years ago. The median age of 9 years in the latest cohort in this study, is comparable with the median age reported in other recent studies [5,30,36]. One explanation for this development could be the decrease in prevalence of the mediastinal lymphoma, a form that is well-known for its younger age of onset in both the current study as in others [6,25,35,37]. In addition, another explanation could be that there seems to be an increase in owners’ willingness to treat elderly cats. In combination with an improvement in veterinary (preventative) specialties and diagnostics, this may have led to an increase in the number of presented older cats. In the present study, there was hardly any difference between patients’ characteristics, including age, of untreated cats and treated cats. With respect to this shift towards older animals, the bimodal distribution in age that was noted in previous studies [6,25,27] disappeared. Since FeLV status could earlier not explain this distribution [35], other possibilities were considered, such as the association with the Siamese breed. The prevalence of Siamese cats among animals with lymphomas has changed during the last decade. Where previous studies recorded 19–33% [5,25,35] Siamese cats, current studies report a prevalence of 2.2–5% [7,8,23]. In the older studies, a high proportion of Siamese cats was associated with a high proportion of mediastinal lymphomas. The fact that in the studies of Court et al. (1997) and Teske et al. (2002) the percentage of mediastinal lymphoma in Siamese and Siamese-related breeds was high, while none of the cats were FeLV positive, suggested a genetic predisposition [6,7,25,35]. In addition, there has been evidence that Siamese cats with lymphoma are generally younger [6]. Results of this study have shown that the decrease in the proportion of Siamese cats might be part of the explanation of both the shift in age, as well as in anatomical forms, but this does not seem to be the only reason. Also, in the non-Siamese cats, there was a significant decrease in mediastinal lymphoma.

In the present study, there was an overrepresentation of male cats, which is comparable to other studies [5,6,24,25,30,35,36]. One study even presented an increased odds ratio of 1.7 for male cats compared to females [8]. Especially in the cohort of Siamese cats, this correlation seems to be more evident [25]. In humans, a similar predisposition has been reported [38,39]. The cause for this sex predisposition has yet to be discovered, but altogether, these results could be suggestive of a hormonal influence. The increase in neutering status seen in this study, on the other hand, might contradict this theory. Chromosomal differences should therefore be considered as another possible cause.

As mentioned before, there seems to be a worldwide shift in the anatomical subtype of feline lymphoma [7,23]. This shift has always been assigned to the change in FeLV/FIV prevalence. In studies from the 1970s and 1980s, about 80% of the cats with malignant lymphoma tested positive for FeLV [3,40], while mediastinal lymphoma was one of the most common forms, accounting for 48% of the cases [41]. Currently, studies from the US, Germany, and the United Kingdom report that due to vaccination programs, less than 14.5% of the lymphomas are FeLV/FIV-associated [7,8,24,36], and these lymphomas are mostly of an alimentary or nasopharyngeal origin. There are, however, arguments that the change in FeLV prevalence might not be such an important factor. Firstly, there are no studies that demonstrate that the decrease in FeLV prevalence in the cat population is associated with a decrease in lymphoma incidence. On the contrary, even increases in lymphoma cases are being reported [7]. Secondly, in countries where FeLV prevalence among cats with lymphoma has always been very low (2–8%), like Australia [27,28] and the Netherlands [25,26,29] where a testing and removal program has been responsible for very low prevalence in the cat population since the early 1980s, the same shift towards alimentary and nasopharyngeal lymphomas is seen. The results of the present study confirm this finding.

Not only the present study, but also several other recent studies have reported an increase in nasopharyngeal and alimentary lymphomas in cats [7,23,29,31,42]. This increase is not only a relative increase, but also an absolute increase [7]. Better diagnostic and reliable options, especially endoscopy, can perhaps partly explain this increase. Both sites, however, are frequently appointed as places where inflammation occurs. It has been suggested that chronic immune stimulation might be an underlying cause of malignant transformation, suggesting that this could be the reason for the high prevalence of nasopharyngeal and alimentary lymphoma in recent studies. This correlation between chronic inflammation and lymphoma has already been demonstrated in humans [3] where a direct association between coeliac disease and lymphoma has been established, as well as a causative role was found for H. pylori in the pathogenesis of gastric MALT lymphoma [43]. A similar correlation has been tried to discover in feline alimentary lymphoma cases, as multiple studies demonstrated that up to 60% of intestinal T-cell lymphoma patients had pre-existing or concurrent chronic clinical illnesses [44,45,46]. However, the histological subtypes in these studies were small cell, low grade intestinal lymphomas, while the reported shift in the current study is for large cell, high grade alimentary lymphomas, including gastric. Of interest is therefore the finding of an association of Helicobacter spp. with large cell gastric lymphoma in cats [47]. A complete explanation for the increase in prevalence of the alimentary and nasopharyngeal lymphomas, however, is still lacking.

Regarding the response rate to the COP-protocol, the 60% CR seen in the present study is comparable to the results seen before in the Netherlands [25,29]. The COP protocol has remained the same over the three time periods. The only modification has been that since 2004 vincristine and cyclophosphamide were administered intraperitoneally instead of intravenously. It was proven that pharmacokinetics, bioavailability, and efficacy were similar between the two ways of administration [29,48]. In other studies worldwide in which the COP-protocol is used, a similar treatment response is seen, varying between 47–72% [6,31,37,49,50]. Some studies added doxorubicin to the protocol. Surprisingly, the complete response rate did not differ [23] or only showed a slight improvement, without affecting survival [23,51]. In the present study, the highest response rates were seen for mediastinal lymphoma, alimentary lymphoma, and miscellaneous lymphoma, including laryngeal, skeletal, and ocular forms (83.3%, 70.4%, and 84.6%, respectively). The results of mediastinal lymphoma are consistent with the results obtained by others [21,38]. The study of Fabrizio et al. (2014), on the other hand, found a lower response rate for this lymphoma type (61.5%). However, an explanation for this could be that a quarter of the patients had involvement of other organs as well, possibly influencing the final outcome [6].

The DFP for alimentary lymphoma in the current study is surprisingly favorable compared to other published results [25,29,51]. Explanations for this difference in response rate were first thought to be related to the suspected difference in survival time between gastric lymphoma and intestinal lymphoma [34], as both of them were included in the alimentary group. However, the current study could not demonstrate a similar beneficial effect on survival time for gastric lymphomas. Another suitable explanation for the increase in DFP for alimentary lymphoma could be the lack of follow-up using diagnostic imaging. In the majority of cases, the response to treatment was based on clinical signs instead of measuring the response on total tumor volume. The intention was to perform regular check-ups by repeated abdominal ultrasonography; however, owner compliance is a necessity for this and was therefore not always feasible. The same applies to the nasopharyngeal lymphomas, in which owners often declined control imaging under anesthesia when the cats seemed to be in remission and doing fine. However, the fact that there are so many longtime survivors in CR in our study with hardly any relapses assumes that the treatment was most of the time sufficient enough.

The median survival time for all cases (274 days) is comparable to or even better than some of the older studies [31,37,50] as well as compared to the results from studies reported in the past in the Netherlands. The median DFP for cases that went into CR in the present study (763 days) was much better than in the two periods before. Especially the nasopharyngeal, mediastinal, and miscellaneous subtypes had a large increase in median DFP with >1357, 1071, and 763 days, respectively. Possible explanations are still lacking.

It is important to mention that survival time depends on multiple risk factors. For example, the MST in the study of Fabrizio et al. (2014) was 484 days [6], which seems to be higher compared to others [31,37,50]. However, this study only included mediastinal lymphoma, which, as mentioned earlier, has a better response to therapy. Differences in anatomical forms included in a study could therefore influence the overall end result. Another strong predictor of the overall response seems to be the FeLV status [37,52]. Differences in FeLV prevalence could eventually influence the overall median survival time. Clear conclusions between the different studies are therefore difficult to make.

## 5. Conclusions

The patient characteristics and anatomic subtypes of cats with malignant lymphoma in the Netherlands have changed over the years. Where previously younger, Siamese cats were involved and mostly mediastinal lymphomas were seen, today the most frequent forms of lymphoma are the nasopharyngeal and alimentary types in cats of an older age. The results of this study suggest that this shift cannot be explained by the changes in FeLV/FIV prevalence. Despite this change in patient characteristics, the overall response rate to chemotherapeutic treatment did not differ. However, the median DFP for cats with a complete response has impressively increased for especially the nasopharyngeal, mediastinal, and miscellaneous subtypes.

## Figures and Tables

**Figure 1 animals-13-02667-f001:**
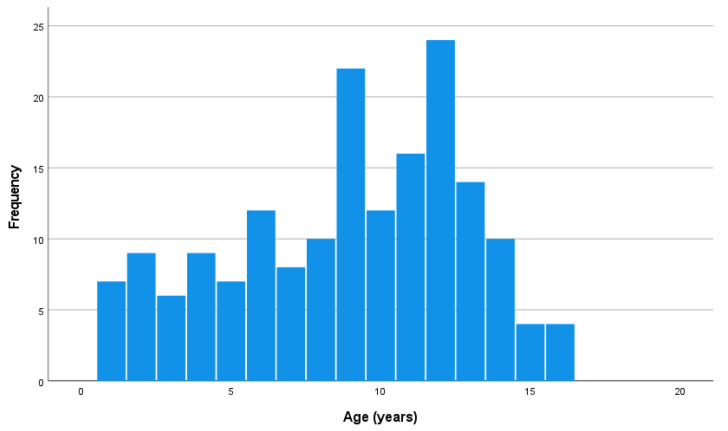
Age distribution of 110 cats treated for malignant lymphoma with cyclophosphamide, vincristine, and prednisolone (COP) at the Utrecht University Clinic for Companion Animals, between 2011 and 2020.

**Figure 2 animals-13-02667-f002:**
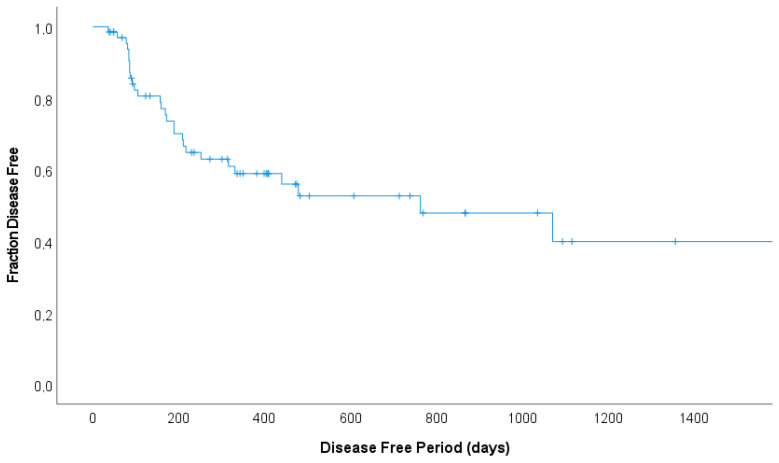
Disease-free survival curve for 66 cats treated according to the COP protocol for malignant lymphoma that went into complete remission (CR). The vertical bars demonstrate censored observations. The median DFP was 763 days with a 1-year estimated DFP of 58.9%, and a 2-year estimated DFP of 52.7%.

**Figure 3 animals-13-02667-f003:**
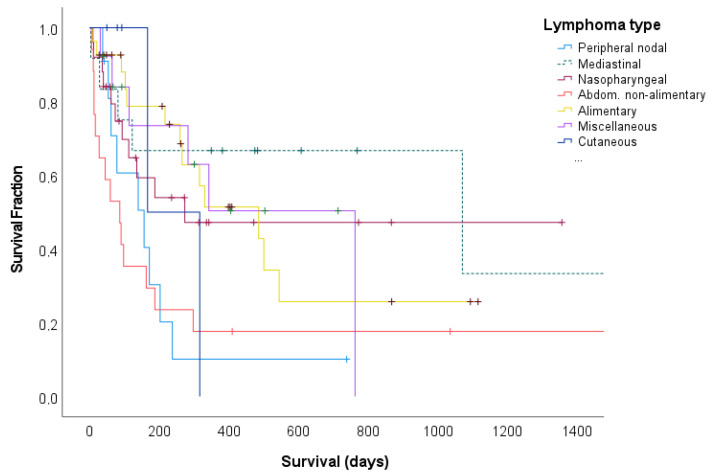
Kaplan–Meier survival curves for 110 cats with malignant lymphoma, categorized according to their anatomical location. The vertical bars demonstrate censored observations. The median survival time was 274 days, with an estimated 1-year survival of 41.3% and a 2-year survival of 34.6%. The 1-year survival rate differed significantly (*p* = 0.019) between the anatomical locations.

**Table 1 animals-13-02667-t001:** Patient characteristics.

Variable	Category	1984–2002		2004–2010		2011–2020 (Treated)		2011–2020 (Untreated)	
		N = 61		N = 26		N = 110		N = 64	
		n	%	n	%	n	%	n	%
Breed	European shorthair	41	67.4	19	73.1	66	60.0	48	75.0
	British shorthair			1	3.8	8	7.3	4	8.9
	Oriental shorthair					4	3.6	2	3.1
	Siamese	16	26.2	1	3.8	8	7.3	3	4.7
	Main Coon					7	6.4	3	4.7
	Norwegian Forest			1	3.8	4	3.6	1	1.5
	Other	4	6.6	4	15.5	13	11.8	3	4.7
Gender	Male intact	10	16.4			2	1.8		
	Male castrated	30	49.2	13	50.0	71	64.5	40	62.5
	Female intact	7	11.5	2	7.7	2	1.8		
	Female castrated	14	23.0	11	42.3	35	31.8	18	37.5
Age	<5 years	20	32.8	4	15.4	22	20.0	9	14.1
	5–10 years	29	47.5	7	26.9	42	38.2	20	31.3
	>10 years	12	19.7	15	57.7	46	41.8	26	54.6
FeLV	Positive	4/54	7.4	0/15		0/61	0.0	1/29	3.4
	Negative	50/54	92.6	15/15		61/61	100	28/29	96.6
FIV	Positive	n.d.		0/15		1/61	1.6	2/29	6.8
	Negative	n.d		15/15		60/61	98.4	27/29	93.2
Anatomical location	Alimentary	11	18.0	3	11.5	27	24.5^a^	20	31.3
	Abdominal			2	7.7	17	15.5	16	25.0
	Mediastinal	22	36.1^a^	1	3.8	12	10.9	2	3.1
	Peripheral nodal	7	11.5	3	11.5	11	10.0	9	14.1
	Nasopharyngeal	8	13.1	10	38.5^a^	25	22.7	16	25.0
	Cutaneous					5	4.6		
	Miscellaneous	13	21.3	7	26.9	13	11.8	1	1.5

^a^ Anatomical location with significant higher prevalence than in other periods.

**Table 2 animals-13-02667-t002:** Response rates and survival data in 110 cats with malignant lymphoma treated with a COP-protocol.

Anatomical Location	n	CR	PR	DFP	Survival
		(n;%)	(n;%)	(days)	(days)
Alimentary	27	19 (70.4)	3 (11.1)	440	486
Abdominal non-alimentary	17	5 (29.4)	4 (23.5)	>410	87
Mediastinal	12	10 (83.3)	1 (8.3)	1071	1071
Peripheral nodal	11	6 (54.5)	3 (27.3)	157	157
Nasopharyngeal	25	15 (60.0)	7 (28.0)	>1357	274
Cutaneous	5	0	4 (80.0)	-	167
Miscellaneous	13	11 (84.6)	1 (7.7)	763	763

**Table 3 animals-13-02667-t003:** Comparison of the median DFP between the different time periods according to their anatomical locations.

Anatomical Location	1984–2002			2004–2010			2011–2020 (Treated)		
	n	CR	Median DFP	n	CR	Median DFP	n	CR	Median DFP
		N (%)	(Days)		N (%)	(Days)		N (%)	(Days)
Total	61	46 (75.4)	251	27	20 (76.9)	421	110	66 (60.0)	763
Alimentary	11	7 (63.6)	245	3	2	228	27	19 (70.4)	486
Abdominal non-alimentary				2	2		17	5 (29.4)	>410
Mediastinal	22	18 (81.8)	251	1	1		12	10 (83.3)	1071
Peripheral nodal	7	6 (85.7)	378	3	2	421	11	6 (54.5)	157
Nasopharyngeal	8	6 (75.0)	358	10	8 (80.0)	388	25	15 (60.0)	>1357
Cutaneous							5	0	-
Miscellaneous	13	9 (69.2)	171	7	5 (71.4)	270	13	11 (84.6)	763

## Data Availability

Not applicable.

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
