# Peer review of "Feline Lymphoma: Patient Characteristics and Response Outcome of the COP-Protocol in Cats with Malignant Lymphoma in The Netherlands"

_animals, 2023, doi:10.3390/ani13162667_

Round 1

Reviewer 1 Report

This study aims to describe characteristics of feline lymphomas in the Netherlands, a country with a low prevalence of FeLV/FIV infection. This project is made up of a large cohort with a large retrospective period of analysis. The aim of this project is interesting, trying to understand why there is a shift in age of patients and anatomical tumor location, independently of FeLV/FIV infection. The paper is well-documented with a special concern in discussing links between epidemiological, clinical characteristics and response to COP treatment. However, this work needs to be improved, in particular at the level of the robustness of the employed method. Firstly, there is no pathological description of the tumors analyzed in this study, either cytologically/histologically (cell morphology, mitoses) or immunohistochemically (T-cell -CD3+- versus B-cell -CD20+- lymphoma, Low- versus High-grade lymphoma -Ki-67-), and the authors are therefore comparing tumors that are not comparable in their biological behavior (High-grade tumors of worse prognosis) and in their response to treatment (high-grade tumors responding better to chemotherapy). Furthermore, almost half of all cats, whether treated (61/110) or untreated (29/64), are not tested for FeLV/FIV infection, which doesn't account for the real link between low prevalence and impact on lymphoma development. In addition, the comparison between treated cats and the whole group is not statistically correct (you can't compare cohort A with cohort B, which contains cohort A) nor is the comparison between the recent cohort and cohorts from previous decades, which don't include the same number of years of study (percentages of breeds per cohort that don't reflect the true annual incidence). Finally, the definition of tumor characteristics and evolution is often approximate, particularly in terms of exact anatomic location (what is abdominal location (internal lymph nodes? liver? spleen?) and in terms of response to treatment, which is not always based on imaging results and even less on cytological/histological analyses. Thus, I recommend publishing this article with major revisions notably with a more robust and accurate method to describe the cohort

Specific comments

Line 46: 100 thousand cats is 100,000 and not 100.000

Line 62-63: Specify new percentages of FeLV+ cats with lymphomas.

Line 105-106: We don’t know if cats treated before 2014 were treated with Teske’s COP modified protocol and if observations are comparable here.

Line 106-8: Specify how many cases were diagnosed by cytology? and by histology? The description of pathological characteristics should be more complete with a morphological description of the tumor and immunohistochemical phenotype (T-cell -CD3+- versus B-cell -CD20+- lymphoma, Low- versus High-grade lymphoma -Ki-67-).

Line 119: Positive FIV case was confirmed by other diagnostic method?

Line 121-122: Add the article reference to justify this use of anatomic classification.

Line 128: Specify when the administration of prednisolone is started.

Line 132: it would be more rigorous to give a dose estimated by weight or surface of cat rather than cat.

Line 135-136: I understand that follow-up is not often easy but you should explain more how many cats were followed only by clinical signs, by imaging or with a cytological analysis at the end of the treatment.

Line 138-139: “Disappearance of clinical signs OR the total volume tumor was considered as a CR”  is it an “or” or an “and”? Because disappearance of clinical signs can be found in a partial response too? Is there no cytology nor biopsy at the end of the treatment?

Line 146: there is too much space between test and to.

Line 149: Be more accurate in the definition of survival. Deaths linked to the studied disease give specific survival whereas deaths, whatever the cause give overall survival. I think that it is specific survival here.

Line 188-189: You describe that there is more Siamese cats in the period 1984-2003 (19 years) compared with the period 2011-2020 (9 years) but no because there is 0.84 siamese cats per year in the first periode (16/19=0.84) and 0.88 per year in the second (8/9=0.88).

Line 204-205 : Same observation for mediastinal lymphomas in 1984-2003 (22/19=1.15 mediastinal lymphomas per year) and in 2011-2020 (12/9=1.33 mediastinal lymphomas per year).

Line 228: If you exclude cats dying of other cause, it is not overall survival but specific survival. Furthermore, survivals are significantly different between several locations but not all. Precise wit a multivariate analysis which are significantly different.

Line 269 : There is not a decrease in the incidence of mediastinal lymphomas (1.15 per year in 1984-2003 versus 1.33 per year in 2011-2020)

Reviewer 2 Report

 Minor editing of English language required

Reviewer 3 Report

In summary: The researchers aimed to study that analyze the distribution in anatomical subtypes in a low FIV/FeLV seroprevalence country, such as the Netherlands, and to determine its consequences on treatment response. In results of this study, researhers stated that patients characteristics of cats with malignant lymphoma in the Netherlands have changed over the years but this cannot be explained by differences in FeLV/FIV prevalence. The researhers also noted that while the overall response rate to treatment has not changed over time, longer disease-free periods (DFPs) have been observed for some lymphoma subtypes compared to 30 years ago.

In manuscript:

In this study, the researchers aimed to evaluate whether this change was also notable in the Netherlands, a country historically known for its low prevalence in FeLV and FIV, and to determine its  consequences on treatment response. To fulfil this aim, they performed a 10-year cohort of 174 cats with lymphoma (110 treated) were included and compared to historical data from previous published reports in the Netherlands. As a result of the study, the researchers obtained the following results; of the 90 cats screened, only one tested positive for FeLV and 3 for FIV. The most current cohort had an increased age (median 8.7 years) and fever Siamese cats (6.3%) compared to previous reports, with alimentary (24.5%) and nasopharyngeal lymphoma (22.7%) being the most common subtypes. The researchers included a total of 174 cats in this study, of which 110 animals were treated with the modified COP-protocol.

In results of this study, researhers stated that despite this change in patient characteristics, the overall response rate to chemotherapeutic treatment did not differ. And also they stated that however, the median disease-free periods (DFP) for cats with a complete response has impressively increased for especially the nasopharyngeal, mediastinal, and miscellaneous subtypes. 

The materials selected and the methods applied by the researchers to realise the stated aim were appropriate and adequate, and detailed information about the methods was given. The results obtained in the study were given in detail, 3 figures and 3 tables were added to make the findings more understandable. The researchers compared their findings with the data in many sources related to the subject of the study and discussed the method they applied for the compare of treatment of feline lymphoma.

Round 2

Reviewer 1 Report

This article, describing changes in feline lymphomas under chemotherapy in a country with a low prevalence of FeLV/FIV infection, has been greatly improved with this revision, with well-argued responses from the authors to justify their choices. The main problems were in the description of the material & methods, and the authors have made the necessary changes to better understand the design of this study and remove the reader's doubts as to the veracity of the statements made. In this sense, I validate the changes and justifications given and I validate the submission of this article in its present form.

Author Response

Dear reviewer #1,

We would like to thank you again for the critical but positive input, which certainly has improved the manuscript.

With kind regards,
Also on behalf of the other authors of the article,

Hannah Versteegh

Reviewer 2 Report

As I previously mentioned, the manuscript describes a scientific study to demonstrate the main clinical presentation of feline lymphoma and the outcome of chemotherapy treatment in the Netherlands. It is well written and presented. The whole study was also well designed and presented interesting results.

Regarding my main concerns, the authors improved the manuscript. However, the Simple Summary remains with some sentences copied from the Abstract. Please avoid repeated sentences. In addition, I strongly suggest to write the main objective in one sentence (and not three paragraphs). All other recommendations were (at least partially) accepted and introduced in the manuscript.

Minor editing.

Author Response

Dear Reviewer #2,

The authors would like to resubmit their manuscript “Feline Lymphoma: Patient Characteristics and Response Out-come to the COP-Protocol in Cats with Malignant Lymphoma in the Netherlands”.

We would like to thank the reviewer for the critical but positive input, which certainly has improved the manuscript. We have rewritten the Simple Summary so overlap with the Abstract is no longer present anymore.

In addition, we have combined the last three paragraphs of the Introduction which describes the objectives of the study and shortened it. We could not get it into one sentence without lost of clarity of text.

As previously mentioned, the global decline in the prevalence of FeLV and FIV may have altered the clinical presentation of feline lymphoma and may therefore have influenced treatment outcomes. The objective of the present study is to test if similar changes in feline anatomical lymphoma subtypes have occurred in the Netherlands, a historically low FeLV prevalence region, as in the rest of the world and if these changes have had an impact on overall treatment outcome. Since this is a single institutional study, any changes observed are unlikely to be related to geographic variations, retroviral status, or variations in treatment protocols.”

With kind regards,
Also on behalf of the other authors of the article,

Hannah Versteegh